# Hierarchical Equivariant Policy via Frame Transfer

**Haibo Zhao** [* 1]  **Dian Wang** [* ✉ 1]  **Yizhe Zhu** [1]  **Xupeng Zhu** [1]  **Owen Howell** [1 2]  **Linfeng Zhao** [1]  **Yaoyao Qian** [1]
**Robin Walters** [1]  **Robert Platt** [1]

{zhao.haib,wang.dian,zhu.xup,howell.o,zhao.linf,qian.ya,r.walters,r.platt}@northeastern.edu

## Abstract

Recent advances in hierarchical policy learning highlight the advantages of decomposing systems into high-level and low-level agents, enabling efficient long-horizon reasoning and precise fine-grained control. However, the interface between these hierarchy levels remains underexplored, and existing hierarchical methods often ignore domain symmetry, resulting in the need for extensive demonstrations to achieve robust performance. To address these issues, we propose Hierarchical Equivariant Policy (HEP), a novel hierarchical policy framework. We propose a frame transfer interface for hierarchical policy learning, which uses the high-level agent's output as a coordinate frame for the low-level agent, providing a strong inductive bias while retaining flexibility. Additionally, we integrate domain symmetries into both levels and theoretically demonstrate the system's overall equivariance. HEP achieves state-of-the-art performance in complex robotic manipulation tasks, demonstrating significant improvements in both simulation and real-world settings. (Code and videos are available at `project page`.)

## 1. Introduction

Learning-based approaches have emerged as a powerful paradigm for developing control policies in sequential decision-making tasks, such as robotic manipulation. By leveraging data-driven methods, policy learning provides a scalable framework for addressing tasks with complex dynamics and high-dimensional observation spaces. Recent advancements in end-to-end policy learning (Zhao et al., 2023; Chi et al., 2023) have shown promising results in

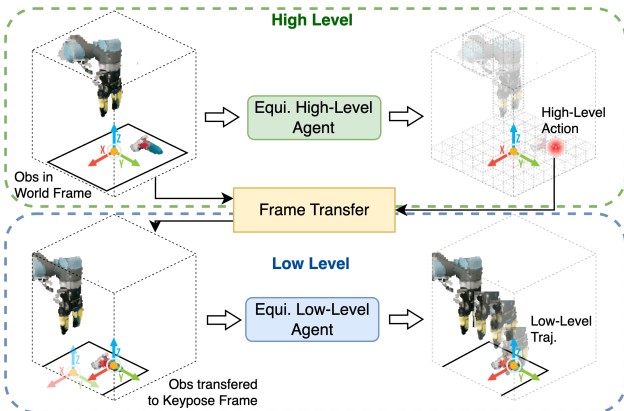

*Figure 1.* **Hierarchical Equivariant Policy (HEP)** is composed of a high-level agent that predicts a coarse translation, a low-level agent that predicts the fine-grained trajectory, and a novel Frame Transfer interface that transfers the coordinate frame of the low-level to the predicted keypose frame from the high-level.

mapping raw sensory inputs to low-level actions such as end-effector trajectories. While these methods exhibit state-of-the-art performance when large amounts of training data are available, they struggle in scenarios with only limited data, due to the large function space required to parameterize complex end-to-end mappings.

A promising alternative strategy is to employ a hierarchical structural prior that decomposes the policy into different levels, e.g., a high-level agent responsible for identifying a goal pose and a low-level agent for trajectory refinement. Hierarchical methods can reduce the complexity of the policy function space by delegating long-horizon reasoning to the high-level module and fine-grained control to the low-level module, enabling efficient learning and execution. Despite their promise, one underexplored question in hierarchical policy learning is what is the right interface between different levels. For example, in robotic manipulation, existing hierarchical methods (Ma et al., 2024; Xian et al., 2023) often impose rigid constraints on the interface between the high-level and low-level agents, where the high-level action is used as the last pose in the low-level trajectory. This constraint limits flexibility and often requires both levels to

---

[*]Equal contribution  [1]Northeastern University  [2]Robotics and AI Institute. Correspondence to: Dian Wang <wang.dian@northeastern.edu>.

*Proceedings of the 42nd International Conference on Machine Learning*, Vancouver, Canada. PMLR 267, 2025. Copyright 2025 by the author(s).

perform fine-grained reasoning in high-dimensional spaces, negating some of the potential benefits of the hierarchical design. Moreover, prior hierarchical methods focus solely on the hierarchical decomposition and do not exploit the domain symmetries often present in robotic tasks, missing an opportunity to further improve generalization and efficiency.

In this paper, we propose a novel hierarchical policy learning framework that overcomes these limitations by introducing a more flexible and efficient interface between the high-level and low-level agents. Specifically, our high-level agent predicts a *keypose* in the form of a coarse 3D location representing a subgoal of the task. This location is then used to construct the coordinate frame for the low-level policy, enabling it to predict trajectories relative to this keypose frame, as shown in Figure 1. This *Frame Transfer* interface maintains a strong inductive bias (by anchoring the low-level policy to a subgoal) yet offers structural flexibility (allowing the low-level policy to refine trajectories locally). Furthermore, Frame Transfer offers a natural fit for integrating domain symmetry by decomposing it into the global symmetry of the subgoal (i.e., the subgoal should transform with the scene) and a local symmetry of the low-level policy (i.e., it should behave consistently in the local keypose frame). By incorporating equivariant structures at both levels, our entire hierarchical system becomes more robust to spatial variations, resulting in significantly improved sample efficiency. Lastly, to better encode 3D sensory information, we adopt a stacked voxel representation (Zhou & Tuzel, 2018), ensuring rich visual features and fast computation.

We summarize our contributions as follows:

- We propose Hierarchical Equivariant Policy (HEP), a novel, sample-efficient hierarchical policy learning framework.

- We introduce Frame Transfer as an interface for hierarchical policy learning, providing effective and flexible policy decomposition.

- We theoretically demonstrate the equivariance of HEP, showing its spatial generalizability. Although equivariance has been used in policy learning, our work is the first to study it in a hierarchical policy.

- We provide a thorough evaluation of our method in both simulation and the real-world. Among 30 RLBench (James et al., 2020) tasks, HEP outperforms state-of-the-art baselines by an average of 10% to 23% in different settings, with particular improvement on tasks requiring fine control or long-horizon reasoning.

## 2. Related Work

**Learning from Demonstrations** (LfD) enables policies to be trained from human demonstrations and generalized to unseen scenarios. One class of LfD learns abstracted keyframe actions (James & Davison, 2022; James et al., 2022; Shridhar et al., 2023; Gervet et al., 2023; Goyal et al., 2023) in terms of the target pose of the gripper, then uses motion planning to interpolate between keyframes. This formulation enables learning with fewer decision steps, but is not suitable for non-prehensile actions like door opening or wiping (Xian et al., 2023; Ma et al., 2024). Another class of LfD mimics the fine-grained trajectory directly (Song et al., 2020; Ye et al., 2022; Toyer et al., 2020; Zhang et al., 2018; Chi et al., 2023; Zhu et al., 2023; Mandlekar et al., 2021; Zhao et al., 2023; Wang et al., 2024), enabling broader task coverage but suffering from overloading the model with details (Zhao et al., 2023), covariant shift (Ke et al., 2021), and poor performance in long-horizon tasks. To bridge these approaches, we introduce Frame Transfer, a novel interface that integrates keyframe-based and trajectory-based models, enhancing flexibility and task adaptability.

**Hierarchical Policy** has been explored for action refinement in a coarse-to-fine manner (Levy et al., 2018; Gualtieri & Platt, 2020; James et al., 2022) or through a two-stage hierarchy for translational and rotational actions (Sharma et al., 2017; Wang et al., 2020; Zhu et al., 2022). While these approaches improve over end-to-end policies, they lack integration of keyframe and trajectory actions. Recent works (Xian et al., 2023; Ma et al., 2024) address this by hierarchically combining a keyframe agent and a trajectory agent, but they fix the goal pose of the trajectory agent with the output from the keyfram agent, limiting flexibility in the low-level and demanding precise reasoning from the high-level agent. In contrast, our framework enables a more adaptable interface between levels, allowing the low-level agent to refine high-level actions.

**Equivariant Robot Learning** leverages geometric symmetries in 3-D Euclidean space, which has emerged as a powerful strategy for boosting the sample efficiency, robustness, and generalisation of robotic-manipulation policies. An expanding line of research now demonstrates that encoding SE(3) equivariance—invariance to translations and in-plane rotations—directly into the perception-to-action pipeline can dramatically reduce the burden of data collection and retraining when scenes or objects are repositioned. Recent works have explored this principle across a wide spectrum of tasks and architectures, from point-cloud networks to image-conditioned diffusion models (Wang et al., 2022; 2021; Liu et al., 2023; Kim et al., 2023; Kohler et al., 2023; Nguyen et al., 2023; 2024; Eisner et al., 2024; Gao et al., 2024). A particularly active subfield studies bi-equivariant pick-and-place policies that remain consistent under simultaneous transformations of both the target object and the robot gripper. Systems in this class—spanning neural descriptors, transporters, and diffusion planners—have achieved state-of-the-art placement accuracy and transfer-

ability (Simeonov et al., 2022; Ryu et al., 2023b;a; Pan et al., 2023; Huang et al., 2022; 2024a;c;b). Complementary research focuses on equivariant grasp synthesis, showing that symmetry-aware grasp networks can generalise to novel object poses with an order of magnitude fewer demonstrations (Zhu et al., 2022; Huang et al., 2023; Hu et al., 2024; Lim et al., 2024). At a finer temporal scale, several groups have begun to embed equivariance into trajectory-generation modules for tasks such as tool use, insertion, or deformable manipulation, reporting sharper motion accuracy and improved long-horizon stability (Jia et al., 2023; Wang et al., 2024; Yang et al., 2024b;a). Unlike these independent applications, we integrate equivariance into a hierarchical policy that unifies keyframe and trajectory-based learning.

# 3. Background

## 3.1. Problem Definition

In this paper, we focus on visuomotor policy learning via Behavior Cloning (BC) in robotic manipulation. We aim to learn a policy $\pi : O \to A$ to map from the observation space $O$ to the action space $A$.

To define the observation and action spaces, let $s = (x, y, z, q, c) \in S = \mathbb{R}^3 \times SO(3) \times \mathbb{R}$ be the space of gripper states where $(x, y, z)$ is a 3D position, $q \in SO(3)$ is an orientation, $c$ is a gripper aperature (open width). The observation space is $o \in O = \mathbb{R}^{n \times (3+k)} \times S^t$ including both a point cloud $P = \{p_i : p_i = (x_i, y_i, z_i, f_i) \in \mathbb{R}^{3+k}\}$ with $k$ dimensional point features (e.g., $k = 3$ for RGB) and $t$ history steps of the gripper state. The action $a = \{a_1, a_2, \ldots, a_m\} \in A = S^m$ contains $m$ control steps of the gripper state.

## 3.2. Equivariance

A function $f$ is equivariant if it commutes with the transformations of a symmetry group $G$, where $\forall g \in G, f(gx) = gf(x)$. This is a mathematical way of expressing that $f$ is symmetric with respect to $G$: if we evaluate $f$ for transformed versions of the same input, we should obtain transformed versions of the same output.

Our objective is to design a policy that is symmetric (equivariant) under the group $g \in T(3) \times SO(2)$, where $T(3)$ represents the group of 3D translations, and $SO(2)$ represents the group of planar rotations around the z-axis of the world coordinate system, $\pi(go) = g\pi(o)$. This symmetry captures the ground truth structure in many robotic tasks without enforcing unnecessary out-of-plane rotation equivariance (which is often invalid due to gravity and the canonical pose of objects).

To define a $T(3) \times SO(2)$ equivariant policy, we first need to define how the group element acts on the observation

and the action. let $g = (t, R_\theta) \in T(3) \times SO(2)$ where $t = (t_x, t_y, t_z)$ and $R_\theta$ is the $2 \times 2$ rotation matrix, $g$ acts on the action $a$ by transforming the gripper pose command. Let $\tilde{R}_\theta = \begin{bmatrix} R_\theta & 0 \\ 0 & 1 \end{bmatrix}$, $ga = \{ga_1, ga_2, \ldots ga_m\}$ where

$$ga_i = (R_\theta(x_i + t_x, y_i + t_y), z_i + t_z, \tilde{R}_\theta \cdot q_i, c_i).$$

$g$ acts on $o$ through transforming the gripper pose in the same way as $a$, and transforming the point cloud $P = \{p_i : p_i = (x_i, y_i, z_i, f_i) \in \mathbb{R}^{3+k}\}$ via $gP = \{gp_i\}$ where

$$gp_i = (R_\theta(x_i + t_x, y_i + t_y), z_i + t_z, f_i).$$

## 3.3. Voxel Maps as Function

In deep learning, voxel maps (3D volumetric data) are typically expressed as tensors. However, it is sometimes convenient to express volumetric data in the form of functions over the 3D space. Specifically, given a one-channel voxel map $V \in \mathbb{R}^{1 \times D \times H \times W}$, we may equivalently express $V$ as a continuous function $\mathcal{V} : \mathbb{R}^3 \to \mathbb{R}$, where $\mathcal{V}(x, y, z)$ describes the intensity value at the continuous world coordinate $(x, y, z)$. Notice that here the domain of $\mathcal{V}$ is the 3D world coordinate frame, not the discrete voxel indices. The relationship between the voxel indices and world coordinates is a linear map defined by the spatial resolution and the origin of the voxel grid.

Similarly, if we have an $m$-channel voxel map $V \in \mathbb{R}^{m \times D \times H \times W}$, we can interpret it as $\mathcal{V} : \mathbb{R}^3 \to \mathbb{R}^m$, where each point $(x, y, z)$ in the volume maps to an $m$-dimensional feature vector. The group $g = (t, \theta) \in T(3) \times SO(2)$ acts on a voxel feature map as

$$(g\mathcal{V})(x, y, z) = \rho(\theta)\mathcal{V}(R_\theta^{-1}(x - t_x, y - t_y), z - t_z), \quad (1)$$

where $t \in T(3)$ acts on $\mathcal{V}$ by translating the voxel location, while $\theta$ acts on $\mathcal{V}$ by both rotating the voxel location and transforming the feature vector via $\rho(\theta) \in GL(m)$, an $m \times m$ invertible matrix known as a group representation.

# 4. Hierarchical Equivariant Policy

The main contribution of our paper is a Hierarchical Equivariant Policy that leverages equivariant learning in both the high-level and low-level agents and employs a novel frame transfer interface to connect them. In this section, we first introduce the overview of our hierarchical policy structure and the novel frame transfer interface. Then, we describe the high-level and low-level agents in detail.

The overview of our system is shown in Figure 2. We factor the policy learning problem into a two-step action prediction using a high-level agent $\pi_{\text{high}}$ and a low-level agent $\pi_{\text{low}}$,

$$\pi(o) = \pi_{\text{low}}(o, t_{\text{high}}) ; t_{\text{high}} = \pi_{\text{high}}(o) \quad (2)$$

where $t_{\text{high}} \in T(3)$ is a 3D translation predicted by $\pi_{\text{high}}$.

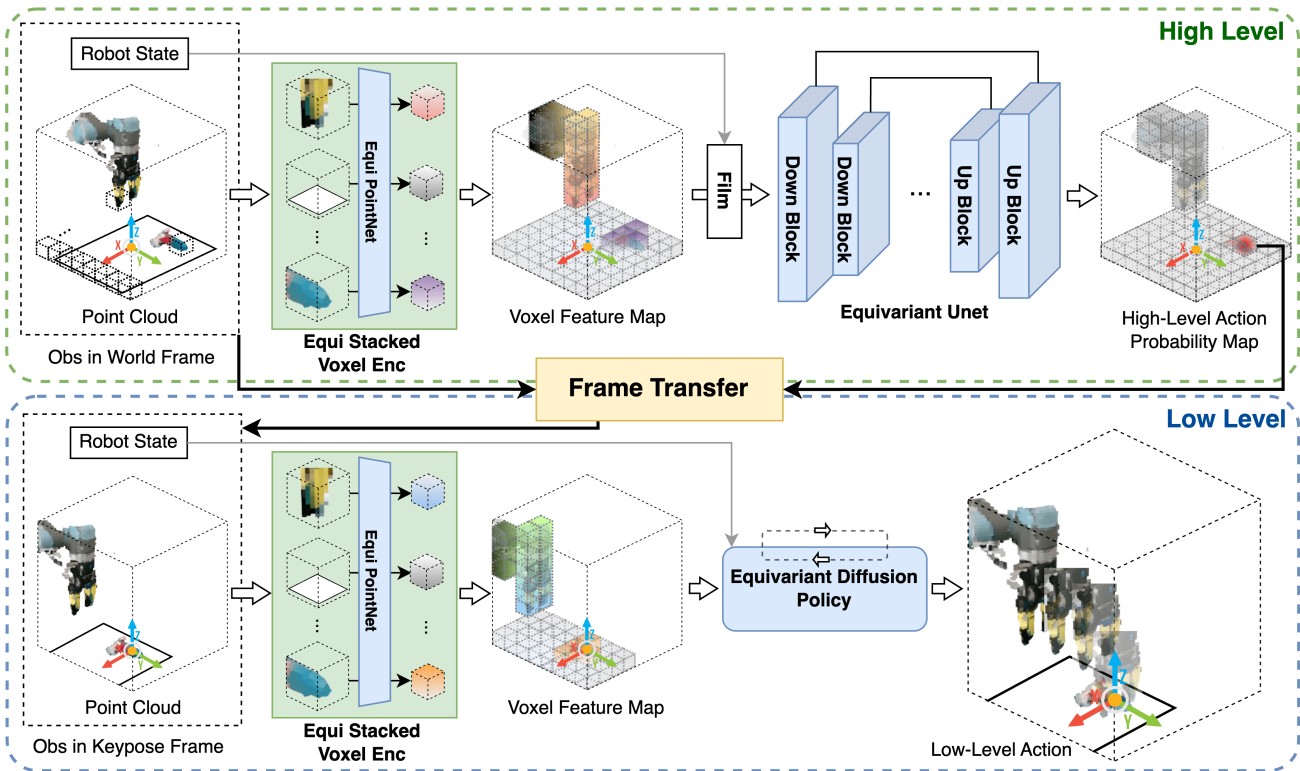

*Figure 2.* **Overview of Hierarchical Equivariant Policy (HEP).** In the highlevel (top), given a point cloud input, we first use an equivariant stacked voxel encoder (green) to process the point cloud and get a voxel feature map. The voxel feature map is then sent to an equivariant UNet (blue) to produce a high-level action probability map. After taking the argmax of the action map as the high-level action, we use Frame Transfer (yellow) to translate the coordinate frame of observation in the low-level (bottom). The translated observation is sent to the stacked voxel encoder (green, same architecture as the one used in the high-level), followed by an equivariant diffusion policy (Wang et al., 2024) (blue) to produce the low-level action.

### 4.1. Frame Transfer Interface

The effectiveness of a hierarchical policy depends largely on the design of the high-level action output and its integration with the low-level agent. Prior approaches (Xian et al., 2023; Ma et al., 2024) often constrain the high-level agent to predict an SE(3) pose, which is then treated as a rigid constraint for the low-level agent by enforcing it as the endpoint of the low-level trajectory. While this design simplifies task decomposition, it restricts flexibility and imposes computational burdens on the high-level agent, which must reason about precise pose constraints in high-dimensional spaces.

To overcome these limitations, we propose a flexible and efficient Frame Transfer interface (Figure 2 middle) between the high- and low-level agents by only passing a $T(3)$ frame rather than constraining the pose. Specifically, our high-level agent predicts a 3D translation $t_{\text{high}}$, which is used as a canonical reference frame for the low-level agent,

$$\pi_{\text{low}}\big(o,\, t_{\text{high}}\big) \;=\; \tau\Big(\phi\big(\tau\big(o,\, t_{\text{high}}\big)\big),\, -t_{\text{high}}\Big), \quad (3)$$

where $t_{\text{high}}$ is the 3D translation (i.e., a keypose) predicted

by the high-level agent, and $\phi$ is a trajectory generator that produces a trajectory based on the transformed observation. $\tau : (O \cup A) \times \mathbb{R}^3 \to O \cup A$ is a Frame Transfer function, which translates the $(x, y, z)$ component of the input observation or action to the input keypose frame. Specifically, we define the $+, -$ operators between $o$ or $a$ and $t_{\text{high}}$ as addition and subtraction on the $(x, y, z)$ component of $o$ or $a$. For example, for $a = (x, y, z, q, c), a + t_{\text{high}} = ((x, y, z) + t_{\text{high}}, q, c)$. The Frame Transfer function $\tau$ is then defined as $\tau(o, t_{\text{high}}) = o - t_{\text{high}}, \tau(a, t_{\text{high}}) = a - t_{\text{high}}$.

The Frame Transfer interface offers several advantages. First, it provides an efficient mechanism that geometrically embeds the high-level action directly into the input of the low-level agent, ensuring seamless communication between the two levels. Second, by representing observations and trajectories in a relative frame, it introduces translation invariance to the low-level agent, simplifying its learning process and improving robustness. Third, unlike prior works (Xian et al., 2023; Ma et al., 2024) which treat the high-level prediction as a rigid motion planning constraint (thus forcing the high-level agent to generate accurate SE(3) poses and

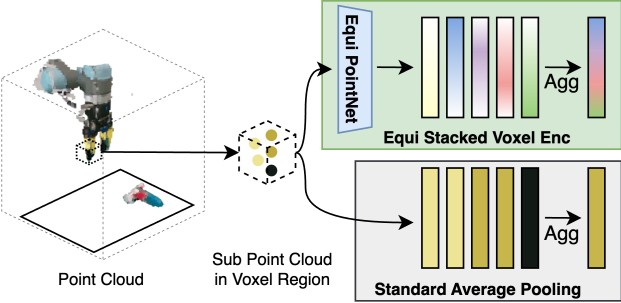

*Figure 3.* **Equivariant Stacked Voxel Encoder.** Compared with the standard average pooling in point cloud voxelization (bottom), stacked voxel representation (top) can provide a richer representation of the points within the region of a voxel.

limiting the policy in an open-loop manner), our approach interprets the high-level output as a flexible constraint. This flexibility reduces the computational burden on the high-level agent, as it only predicts a 3D translation, while preserving the system's capability to operate in both open-loop and closed-loop control settings.

### 4.2. High-level Agent

To efficiently predict the high-level action $t_{\text{high}} \in T(3)$, we represent it as a voxel map discretizing $\mathcal{V}_a : \mathbb{R}^3 \to \mathbb{R}$ where $\mathcal{V}_a(t)$ represents the probability of translation $t$ (see subsection 3.3) . This provides a dense spatial representation and naturally handles translation multi-modality (Shridhar et al., 2023). The center of the voxel with the highest predicted probability is then selected as the high-level agent's final output, $t_{\text{high}} = \arg\max \mathcal{V}_a$. Accordingly, the input observation is voxelized to $\mathcal{V}_o : \mathbb{R}^3 \to \mathbb{R}^3$ (where the output of $\mathcal{V}_o$ is RGB), and we use an SO(2)-equivariant 3D U-Net $\psi : \mathcal{V}_o \to \mathcal{V}_a$ to enforce $g \in T(3) \times \mathrm{SO}(2)$ symmetry, $\psi(g\mathcal{V}_o) = g\psi(\mathcal{V}_o)$. The entire high-level structure is shown in Figure 2 top.

During training, the high-level agent's objective is to minimize the discrepancy between its predicted voxel heatmap $\mathcal{V}_a$ and the ground truth one-hot heatmap $\mathcal{V}_a^*$, derived from expert demonstrations, using the cross-entropy loss,

$$\mathcal{L}_{\text{high}} = - \sum_{x,y,z} \mathcal{V}_a^*(x,y,z) \log(\hat{\mathcal{V}}_a(x,y,z)), \quad (4)$$

where $\hat{\mathcal{V}}_a(x,y,z)$ is the probability for voxel $(x,y,z)$ obtained by applying a softmax over the predicted heatmap.

### 4.3. Stacked Voxel Representation

As our high-level agent uses 3D voxel grids as the visual input, the voxel encoder plays a crucial role in the policy. Standard 3D convolutional encoders impose a heavy computational burden, which often requires aggressive resolution

compression that reduces the fine details in the observation. To address this limitation, we adopt Stacked Voxels (Zhou & Tuzel, 2018) from the 3D vision literature, which preserve fine-grained spatial cues by replacing voxel downsampling with a PointNet (Qi et al., 2017) that aggregates information from all points within the spatial extent of each voxel.

Specifically, given a point cloud $P$, we first partition it into $H \times W \times D$ point sets, where each set $P_j \subseteq P$ corresponds to the points contained within a voxel $j$ in the $H \times W \times D$ voxel grid. Each point set $P_j$ is processed by an equivariant PointNet $l : P_j \mapsto \mathcal{V}(j_x, j_y, j_z)$ to produce a $c$-dimensional aggregated feature vector for the voxel $j$. Repeating for all voxels results in a voxel grid feature map with dimensions $c \times H \times W \times D$. This feature map is then used as input to subsequent 3D convolutional networks.

This process, illustrated in Figure 3, retains more nuanced shape information compared to simple voxel downsampling. Moreover, we prove that the stacked voxel representation maintains equivariance. (See proof in Appendix C.)

**Proposition 4.1.** *For $g = (t, \theta) \in T(3) \times \mathrm{SO}(2)$, if the PointNet $l$ is $\mathrm{SO}(2)$-equivariant and $T(3)$-invariant, i.e., $l(gP_j) = \rho(\theta)l(P_j)$, then the stacked voxel representation $\nu : P \mapsto \mathcal{V}$ s.t. $\nu(P)(j_x, j_y, j_z) = l(P_j)$ is $\mathrm{T}(3) \times \mathrm{SO}(2)$-equivariant, i.e., $\nu(gP) = g\nu(P)$.*

In practice, we implement the $T(3)$-invariance in the Point-Net by using the relative position to the center of each voxel, and implement the $\mathrm{SO}(2)$-equivariance using escnn (Cesa et al., 2022).

### 4.4. Low-level Agent

After predicting the high-level action $t_{\text{high}}$ and using Frame Transfer to canonicalize the observation, our low-level trajectory generator $\phi$ needs to create an $\mathrm{SE}(3)$ trajectory for the robot gripper. As shown in Figure 2 bottom, we first process the observation with a stacked voxel encoder, then leverage an equivariant diffusion policy (Wang et al., 2024) to represent the policy $\phi$, which denoises the trajectory from a randomly sampled noisy trajectory. Specifically, we model a conditional noise prediction function $\varepsilon : o, a^k, k \mapsto e^k$, where the observation $o$ is the denoising conditioning, $a^k$ is a noisy action, $k$ is the denoising step, and $e^k$ is the predicted noise in $a^k$ s.t. the noise-free action $a = a^k - e^k$. The model $\varepsilon$ is implemented as an $\mathrm{SO}(2)$-equivariant function, $\varepsilon(go, ga^k, k) = g\varepsilon(o, a^k, k)$, to ensure the policy $\phi$ it represents is $\mathrm{SO}(2)$-equivariant, $\phi(go) = g\phi(o)$. See (Wang et al., 2024) for more details.

During training, given an expert observation trajectory pair $(o, a)$, we first use the translation $t_n$ from the last step $a_n$ as the keypose, then apply frame transfer to get

$o^* = \tau(o, t_n), a^* = \tau(a, t_n)$. The low-level loss is

$$\mathcal{L}_{\text{low}} = \left\| \varepsilon(o^*, a^* + e^k, k) - e^k \right\|^2, \qquad (5)$$

where $e^k$ is a random noise conditioned on a randomly sampled denoising step $k$.

### 4.5. Symmetry of Policy

In this section, we describe the overall $T(3) \times SO(2)$ symmetry of our hierarchical architecture. As is shown in Figure 4, a transformation in the observation should lead to the same transformation in both levels of HEP. Specifically, we decompose the symmetry into a rotation and translation, and prove each separately.

Let $\pi$ be a hierarchical policy composed of a high-level agent $\pi_{\text{high}}$, a low-level agent $\pi_{\text{low}}$, and frame-transfer functions $\tau$ (see section 4).

**Proposition 4.2** (Hierarchical $SO(2)$ Equivariance). *$\pi$ is $SO(2)$-equivariant when the following assumptions hold for $g \in SO(2)$:*

1. *The high-level policy $\pi_{high}$ is $SO(2)$-equivariant, $\pi_{high}(go) = g\pi_{high}(o)$*

2. *The low-level policy $\pi_{low}$ is $SO(2)$-equivariant, $\pi_{low}(go, gt_{high}) = g \cdot \pi_{low}(o, t_{high})$*

3. *The Frame Transfer function $\tau$ is $SO(2)$-equivariant.*

In Appendix A we show that the *entire* hierarchical policy $\pi$ is $SO(2)$-equivariant so that rotating the observation $o$ results in an action rotated in the same way.

**Proposition 4.3** (Hierarchical $T(3)$ Equivariance). *$\pi$ is $T(3)$-equivariant when the following assumptions hold for $t \in T(3)$*

1. *$\pi_{high}$ is $T(3)$-equivariant, $\pi_{high}(o + t) = t + \pi_{high}(o)$*

2. *The Frame Transfer function $\tau$ is $T(3)$-invariant, and satisfies $\tau(o, t_{high}) = \tau(o + t, t_{high} + t)$*

Notably, even if the low-level policy $\pi_{\text{low}}$ is not $T(3)$-equivariant, the *entire* hierarchical policy $\pi$ is $T(3)$-equivariant. This is proven in Appendix B.

## 5. Simulation Experiment

### 5.1. Experimental Settings

To evaluate our policy, we first perform experiments in simulated environments in the RLBench (James et al., 2020) benchmark implemented using CoppeliaSim (Rohmer et al., 2013) and PyRep (James et al., 2019). The simulated environments contain a 7-joint Franka Panda robot equipped

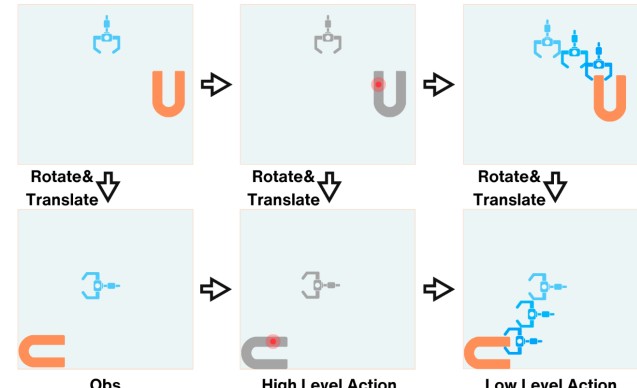

*Figure 4.* **Equivariance in HEP.** When the observation is rotated and translated, the high- and low- level actions are rotated and translated accordingly.

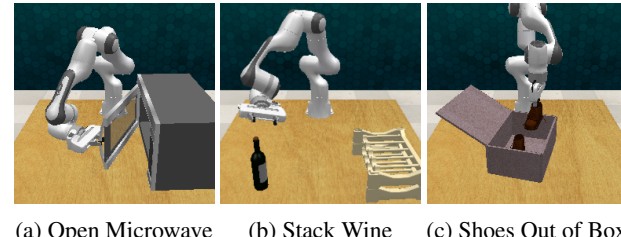

(a) Open Microwave    (b) Stack Wine    (c) Shoes Out of Box

*Figure 5.* **The Simulation Tasks from RLBench (James et al., 2020).** See Appendix F for all environments.

with a parallel gripper, as well as four RGB-D cameras to provide the point cloud observation.

We evaluate our model on 30 RLBench tasks, among which 20 are widely used in the prior works like (Xian et al., 2023). The remaining 10 are challenging tasks that demand precise control, such as Lamp On, or long-horizon planning, like Push 3 Buttons. A subset of the 30 simulation tasks is shown in Figure 5. Each task is trained using 100 demonstrations, more detailed task descriptions and visualizations are provided in Appendix F.

We consider two different control settings, open-loop and closed-loop control. In closed-loop, we use each control step in the dataset as the low-level's target, and next keyframe is used as the label for the high-level agent. In open-loop, we use the keyframe (i.e., some key actions in the entire trajectory like pick, place, etc.) defined by the prior work (Shridhar et al., 2023) as the target for the high-level agent, then construct the low-level target by interpolating between the consecutive keyframes. In principle, the open-loop setting requires fewer prediction steps to finish a task, while the closed-loop setting makes the policy more responsive. Thanks to the flexibility of our Frame Transfer interface, our policy can operate in both settings, while some prior works are limited in the open-loop setting.

*Table 1.* **Performance of Different Models Across Various Tasks in Simulation.** Success rates (in percentages) are reported for each task. Bolded values indicate the best performance for each task, and improvements are shown in blue where applicable.

| Method (Open-loop) | Mean | Pick/Lift | Push Button | Knife on Board | Put Money | Reach Target | Slide Block | Stack Wine | Take Money | Take Umbrella | Pick up Cup |
|---|---|---|---|---|---|---|---|---|---|---|---|
| Ours (HEP) | **88**(+10) | **99**(+1) | **100**(+1) | **96**(+5) | 98(-1) | **100** | **100**(+2) | **100**(+7) | 90(-10) | **100**(+1) | **98**(+4) |
| Chained Diffuser | 78 | 98 | 96 | 91 | **99** | **100** | 98 | 93 | **100** | 96 | 94 |
| 3D Diffuser Actor | 56 | 98 | 99 | 84 | 88 | **100** | 98 | 90 | 89 | 99 | 94 |

| Method (Open-loop) | | Unplug Charger | Close Door | Open Box | Open Fridge | Frame off Hanger | Open Oven | Books on Shelf | Wipe Desk | Cup in Cabinet | Shoe out of Box |
|---|---|---|---|---|---|---|---|---|---|---|---|
| Ours (HEP) | | **99**(+4) | **90**(+24) | **100**(+4) | **83**(+15) | **93**(+8) | **87**(+1) | **99**(+7) | **77**(+12) | **76**(+8) | **90**(+12) |
| Chained Diffuser | | 95 | 76 | 96 | 68 | 85 | 86 | 92 | 65 | 68 | 78 |
| 3D Diffuser Actor | | 49 | 7 | 15 | 41 | 71 | 3 | 36 | 5 | 1 | 21 |

| Method (Open-loop) | | Open Microwave | Turn on Lamp | Open Grill | Stack Blocks | Stack Cups | Push 3 Buttons | USB in Computer | Open Drawer | Put Item in Drawer | Sort Shape |
|---|---|---|---|---|---|---|---|---|---|---|---|
| Ours (HEP) | | **82**(+26) | **95**(+55) | **99**(+4) | **54**(+4) | **32**(+4) | **99**(+12) | **90**(+16) | **94**(+10) | **95**(+7) | **22**(+3) |
| Chained Diffuser | | 56 | 40 | 95 | 10 | 12 | 86 | 74 | 84 | 88 | 10 |
| 3D Diffuser Actor | | 46 | 20 | 70 | 50 | 28 | 87 | 42 | 71 | 70 | 19 |

| Method (Closed-loop) | Mean | Turn On Lamp | Open Microwave | Push 3 Buttons | Open Drawer | Put Item in Drawer | Slide Block | Stack Wine | Take Money | Take Umbrella | Pick up Cup |
|---|---|---|---|---|---|---|---|---|---|---|---|
| Ours (HEP) | **79**(+22) | **60**(+32) | **64**(+22) | **37**(+36) | **95**(+41) | **76**(+28) | **95**(+20) | **89**(+10) | **94**(+14) | **90**(+9) | **93**(+15) |
| EquiDiff | 57 | 28 | 42 | 1 | 54 | 48 | 75 | 79 | 80 | 81 | 78 |

Time

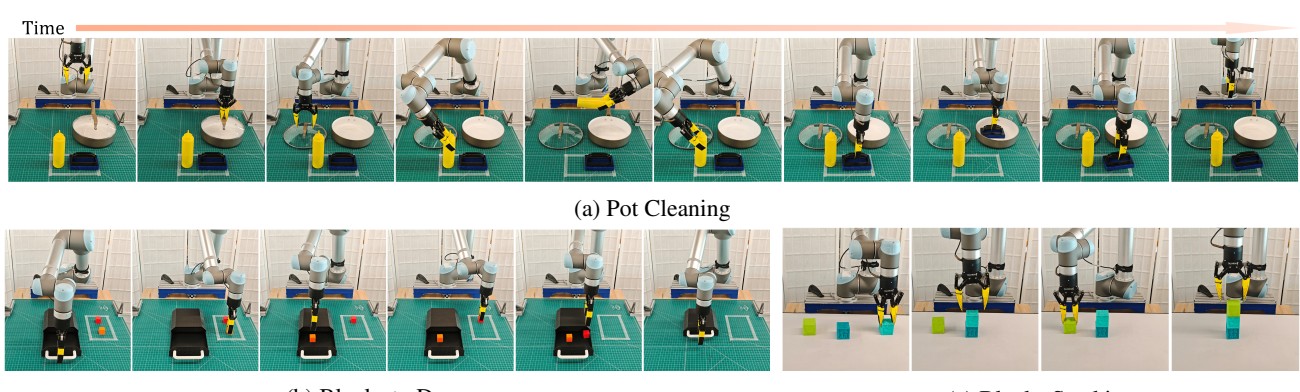

(a) Pot Cleaning

(b) Blocks to Drawer

(c) Blocks Stacking

*Figure 6.* **Real-world Experiment Setting.** Figure 6a: Pot cleaning, the robot needs to open the pot lid, pour detergent into the pot, and clean it with a sponge. Figure 6b: Blocks to drawer, the robot needs to open the drawer, place two blocks inside, and close the drawer. Figure 6c: Blocks stacking, the robot needs to stack three blocks one by one.

## 5.2. Baseline

We compare our method against the following baselines. **3D Diffuser Actor**: an open-loop agent that combines diffusion policies (Chi et al., 2023) with 3D scene representations. **Chained Diffuser**: an open-loop hierarchical agent that uses Act3D (Gervet et al., 2023) in the high-level and diffusion policy in the low-level. **Equivariant Diffusion Policy (EquiDiff)**: an SO(2)-equivariant, closed-loop policy that applies equivariant denoising.

## 5.3. Results

Table 1 presents the comparison in terms of the evaluation success rates of the last checkpoint across 100 trials.

**Open-loop Results:** Our model outperforms the baselines in 28 out of the 30 tasks, achieving an average absolute

improvement of **10%**. The task where HEP falls short of achieving the best results is Take Money. Further investigation reveals that HEP achieves 98% success rate at earlier checkpoints but fails at the final checkpoint, likely due to overfitting. Tasks involving precise actions or long-horizon trajectories e.g., Lamp-on and Push 3 Buttons also exhibited consistently high success rates, demonstrating the adaptability of our method to diverse task requirements. We also compare our model with hierarchical diffusion policy (Ma et al., 2024) in Appendix H

**Closed-loop Results:** Here we consider 10 selected tasks that represent the full diversity and complexity of the complete task set. The closed-loop setting requires longer-horizon trajectories, making it harder to succeed in evaluation. Despite this, our model consistently outperforms EquiDiff across all 10 tasks, achieving an average absolute

improvement of **23%**. This improvement underscores the effectiveness of HEP in handling the increased complexity of long-horizon decision-making.

### 5.4. Ablation Study

To validate the impact of our contributions,we perform an ablation study in six tasks considering the following configurations: **No Hierarchy**: removes high-level agent and uses low-level agent only. **No Equi**: same architecture but removes all equivariant structure. **No Stacked Voxel**: removes the stacked voxel encoder. **No FT**: removes the Frame Transfer interface and uses the high-level action as an additional conditioning in the low-level. **No Equi No FT**: combination of No Equi and No FT.

*Table 2.* **Ablation Study Results.**

| Method | Mean |
|---|---|
| No Hierarchy | 0.51 |
| No Equi No FT | 0.60 |
| No Equi | 0.70 |
| No FT | 0.78 |
| No Stacked Voxel | 0.84 |
| **Complete Model** | **0.94** |

As is shown in Table 2, removing equivariance makes the most significant negative impact on our model, reducing the mean success rate by 24%. The performance drop when removing Frame Transfer and stacked voxel encoder is 16% and 10%, respectively, demonstrating the importance of all three key pieces of our model. Moreover, the 10% performance difference between No Equi and No Equi No FT shows the potential of Frame Transfer beyond our model. See Table 7 in the Appendix for the full table.

## 6. Real-World Experiment

In this section, we evaluate our method on a real robot system comprised of a UR5 robot and 3 Intel Realsense (Keselman et al., 2017) D455 RGBD sensors. Details on the experiment setting are given in Appendix G.

**Baseline Comparison**   We experiment in three tasks as shown in Figure 6. These tasks are challenging due to their extreme long horizon (can be divided into 6 to 9 sub-tasks) and the diverse types of manipulation involved. Evaluations are conducted in 20 trials: 10 with object placements similar to the training dataset's and 10 with unseen placements.

As shown in Table 3, our model successfully completes the tasks under open-loop control. Most failures occur due to the slight misalignment between the gripper and the object, likely caused by poor depth quality of the sensors. We further evaluate our model in a closed-loop setting, where it achieves similar performance to the open-loop version in two of the three tasks. However, in Pot Cleaning, while the agent progresses further in the task, it becomes stuck in a recurrent cleaning loop. This likely results from the lack of history information in the observations, preventing the agent from recognizing when to exit the cleaning phase. In

*Table 3.* **Performance of Different Models in the Real-World.**

| Task | Pot Cleaning | Blocks to Drawer | Blocks Stacking |
|---|---|---|---|
| Number of Demo | 30 | 20 | 30 |
| Chained Diffuser | 0.3 | 0.2 | 0.4 |
| Open-loop HEP (Ours) | **0.8** | **0.85** | **0.9** |
| Closed-loop HEP (Ours) | - | **0.8** | **0.9** |

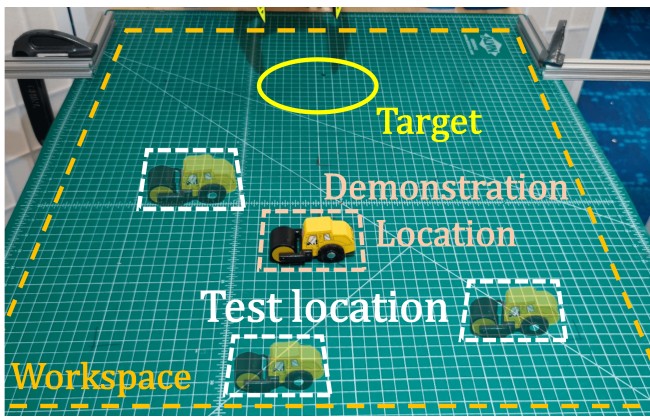

*Figure 7.* **One-Shot Test.** The model is trained on a single demonstration to evaluate its generalization capability.

contrast, the open-loop version follows a single keypose for cleaning, facilitating a smoother transition to the next stage.

**One-Shot Generalization**   To evaluate the generalizability of our model, we perform a one-shot experiment where the model is trained to finish a pick-place task with only one demonstration. During testing, the object is placed in unseen poses, as shown in Figure 7. The results in Table 4 demonstrate the strong generalizability of our model, achieving an 80% success rate over 20 trials. For comparison, we evaluate Chained Diffuser under the same setting, but it only succeeded when the toy car was positioned exactly as in the demonstration. This result highlights the superior generalization ability of our approach, enabling robust execution of manipulation tasks from limited training data.

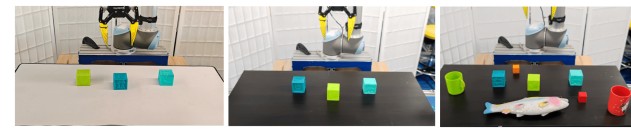

*Figure 8.* **Environment Variations.** Left shows the training environment. Middle and right are the test environment with variation.

**Robust to Environment Variations**   In this experiment, we evaluate the robustness of our trained model under environmental variations. Specifically, we introduced modifications to the Block Stacking task during test time by changing the color of the table (Color) and additionally adding un-

*Table 4.* **Results of One-Shot Generalization Experiment.**

| Model | Success Rate |
|---|---|
| Chained Diffuser | 0.05 |
| HEP (Ours) | **0.80** |

*Table 5.* **Results of Environmental Variation Experiment.**

| Method | No Variation | Color | Color+Objects |
|---|---|---|---|
| Chained Diffuser | 0.4 | 0 | 0 |
| HEP (Ours) | **0.9** | **0.9** | **0.6** |

related objects as distractors (Color+Objects), as shown in Figure 8. The result is shown in Table 5. Surprisingly, our model demonstrates exceptional adaptability, achieving 90% and 60% success rate under those two test-time variations, whereas the baseline fails to complete the task with those distractions.

## 7. Conclusion

In this work, we propose an Hierarchical Equivariant Policy for visuomotor policy learning. By utilizing Frame Transfer, our architecture naturally has both translational and rotational equivariance. Experimentally, HEP achieves significantly higher performance than previous methods on behavior cloning tasks that require fine motor control. While our work provides a solid foundation for hierarchical policies with geometric structure, several future directions remain open for exploration. One key limitation is that our experiments focus on tabletop manipulation. Extending HEP to more complex robotic tasks, such as humanoid motion, is a promising direction. Another limitation is the lack of memory mechanisms, which can be critical for tasks requiring history information. Future work could explore integrating Transformers (Vaswani, 2017) to enhance temporal reasoning. Finally, expanding Frame Transfer to incorporate both translational and rotational specification could further improve the effectiveness of hierarchical policies.

## Acknowledgment

This work was supported in part by the National Science Foundation under Grant Ns. 1750649, 2107256, 2134178, 2314182, and 2409351, and by NASA under Grant No. 80NSSC19K1474. The authors would like to extend their gratitude to Boce Hu for designing the fin-ray gripper fingers and for proofreading the paper, to Heng Tian for designing and fabricating components for the robot experiments, and to Shaoming Li for collecting demonstrations for the robot experiments. We gratefully acknowledge their support. Owen L. Howell is grateful to the National Science Foundation Graduate Research Fellowship Program (NSF-GRFP) for financial support.

## Impact Statement

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

# A. Proof: The Full Policy is $SO(2)$ **Equivariant**

Let us prove that the policy is $SO(2)$ Equivariant and satisfies

$$\forall g \in SO(2), \quad \pi(g \cdot o) = g \cdot \pi(o) \tag{6}$$

We will prove this in two steps.

## A.1. Low-level Equivariance

First, let us prove that the low-level agent is $SO(2)$ equivariant. The low-level policy can be written as

$$\pi_{low}(o, t_{high}) = \tau(\phi(\tau(o, t_{high})), -t_{high})$$

The frame transfer functions satisfy

$$\forall g \in SO(2), \quad \tau(g \cdot o, g \cdot t) = g \cdot \tau(o, t)$$

and the diffusion policy satisfies

$$\forall g \in SO(2), \quad \phi(g \cdot o) = g \cdot \phi(o)$$

Thus, we have that

$$\forall g \in SO(2), \pi_{low}(g \cdot o, g \cdot t_{high}) = \tau(\phi(\tau(g \cdot o, g \cdot t_{high})), g \cdot -t_{high})$$

Using the frame transfer function property $\tau(g \cdot o, g \cdot t) = g \cdot \tau(o, t))$ we have that

$$\forall g \in SO(2), \pi_{low}(g \cdot o, g \cdot t_{high}) = \tau(\phi(g \cdot \tau(o, t_{high})), g \cdot -t_{high})$$

Using the $SO(2)$ equivariance of the diffusion policy and the properties of the frame transfer functions, we have that

$$\forall g \in SO(2), \tau(\phi(g \cdot \tau(o, t_{high})), g \cdot -t_{high}) = \tau(g \cdot \phi(\tau(o, t_{high})), g \cdot -t_{high}) = g \cdot \tau(\phi(\tau(o, t_{high})), -t_{high})$$

Thus,

$$\forall g \in SO(2), \pi_{low}(g \cdot o, g \cdot t_{high}) = g \cdot \tau(\phi(\tau(o, t_{high})), -t_{high})$$

And by definition,

$$\tau(\phi(\tau(o, t_{high})), -t_{high}) = \pi_{low}(o, t_{high})$$

Thus, we have that

$$\forall g \in SO(2), \pi_{low}(g \cdot o, g \cdot t_{high}) = g \cdot \pi_{low}(o, t_{high})$$

$\square$

## A.2. Full Policy Equivariance

Using the equivariance of the low-level policy, let us show that the full policy is $SO(2)$ equivariant. The high-level, low-level and diffusion policies satisfy

$$\forall g \in SO(2), \quad \pi_{high}(g \cdot o) = g \cdot \pi_{high}(o)$$
$$\forall g \in SO(2), \quad \pi_{low}(g \cdot o, g \cdot t_{high}) = g \cdot \pi_{low}(o, t_{high})$$

Now, combining high-level and low-level policy together we got that:

$$\pi(o) = \pi_{low}(\pi_{high}(o), o)$$

When g acting on both the input observation, we have that

$$\forall g \in SO(2), \quad \pi(g \cdot o) = \pi_{low}(\pi_{high}(g \cdot o), g \cdot o)$$

Now, via the $SO(2)$ equivariance of the high-level policy $\pi_{high}(g \cdot o) = g \cdot \pi_{high}(o)$ we have that

$$\forall g \in SO(2), \pi_{low}(\pi_{high}(g \cdot o), g \cdot o) = \pi_{low}(g \cdot \pi_{high}(o), g \cdot o)$$

Thus, using the $SO(2)$ equivariance of the low-level policy $\pi_{low}(g \cdot \pi_{high}(o), g \cdot o) = g \cdot \pi_{low}(\pi_{high}(o), o)$ we have that

$$\forall g \in SO(2), \quad \pi(g \cdot o) = g \cdot \pi_{low}(\pi_{high}(o), o)$$

Note that the $\pi_{low}(\pi_{high}(o), o)$ is just the expression for $\pi(o)$. Thus, we must have that

$$\forall g \in SO(2), \quad \pi(g \cdot o) = g \cdot \pi(o)$$

holds. □

## B. Proof: The Full Policy is $T(3)$ Equivariant

As defined in subsection 4.1, $+, -$ as operators between $o$ or $a$ and $t_{\text{high}}$ as addition and subtraction on the $(x, y, z)$ component of $o$ or $a$. Similarly, we can define the translation $t \in T(3)$ acting on $o$ or $a$ as an addition to the $(x, y, z)$ component as $o + t$ or $a + t$.

First, suppose that the high-level policy $\pi_{high}(o)$ is $T(3)$-equivariant so that

$$\forall t \in T(3), \quad \pi_{high}(o + t) = t + \pi_{high}(o)$$

which is simply the statement that shifting the scene shifts the high-level policy in the same way. Now, we will show that the full hierarchical policy satisfies the equivariance condition. The low-level policy is determined by

$$\pi_{low}(\pi_{high}(o), o) = \tau(\phi(\tau(o, \pi_{high}(o))), -\pi_{high}(o))$$

How does the low-level policy transform under a translation? Using $\pi(o) = \pi_{low}(\pi_{high}(o), o)$, we have that

$$\pi(o + t) = \pi_{low}(o + t, \pi_{high}(o + t))$$

Using the definition of the low-level policy, we have that

$$\pi(o + t) = \pi_{low}(o + t, \pi_{high}(o + t)) = \tau(\phi(\tau(o + t, \pi_{high}(o + t))), -\pi_{high}(o + t))$$

Now, using the equivariance of high-level policy, we have that $\pi_{high}(o + t) = t + \pi_{high}(o)$ so that

$$\tau(\phi(\tau(o + t, \pi_{high}(o + t)), -\pi_{high}(o + t)) = \tau(\phi(\tau(o + t, \pi_{high}(o) + t), -\pi_{high}(o) - t)$$

Now, we can simplify this expression via the fact that the frame transfer $\tau$ function is $T(3)$ invariant. We have that $\tau(o + t, \pi_{high}(o) + t) = \tau(o, t_{high})$ which implies that

$$\tau(\phi(\tau(o + t, \pi_{high}(o) + t), -\pi_{high}(o) + t) = \tau(\phi(\tau(o, \pi_{high}(o)), -\pi_{high}(o) - t)$$

Now, note that $\tau(o + t, \pi_{high}(o) + t) = \tau(o, t_{high})$ implies that $\tau(o, -\pi_{high}(o) - t) = \tau(o + t, -\pi_{high}(o))$. Using the fact that $\tau(o + t, -\pi_{high}(o)) = \tau(o, -\pi_{high}(o)) + t$, we have that

$$\tau(\phi(\tau(o, \pi_{high}(o)), -\pi_{high}(o) - t) = \tau(\phi(\tau(o, \pi_{high}(o)), -\pi_{high}(o)) + t$$

Thus, combining the above expressions and using the definition of the low-level policy, we have that

$$\pi_{low}(o + t, \pi_{high}(o + t)) = \tau(\phi(\tau(o, \pi_{high}(o)), -\pi_{high}(o) - t) = t + \tau(\phi(\tau(o, \pi_{high}(o)), -\pi_{high}(o)) = t + \pi_{low}(o, \pi_{high}(o))$$

Thus, we have that

$$\pi_{low}(o + t, \pi_{high}(o + t)) = t + \pi_{low}(o, \pi_{high}(o))$$

This is just the expression for the full policy function as $\pi(o) = \pi_{low}(o, \pi_{high}(o))$. Ergo, we must have that

$$\pi(o + t) = t + \pi(o)$$

and the full policy is $T(3)$-Equivariant. □

## C. Proof: Equivariance of the Stacked Voxel Representation

*Proof.* We aim to prove that the stacked voxel representation $\nu$ is $T(3) \times SO(2)$-equivariant, i.e.,

$$\nu(gP) = g\nu(P),$$

where $g \in T(3) \times SO(2)$ is a group transformation.

Define a point set selection function $m : j_x, j_y, j_z, P \mapsto P_J$ that selects the subset of points $P_j \subseteq P$ within the voxel indexed by $(j_x, j_y, j_z)$. By the definition, $m$ is an equivariant function $m(g(j_x, j_y, j_z), gP) = gm(j_x, j_y, j_z, P)$.

The stacked voxel representation $\nu$ for a given voxel location $j_x, j_y, j_z$ can be written as:

$$\nu(P)(j_x, j_y, j_z) = l\left(m(j_x, j_y, j_z, P)\right),$$

where:

- $m(j_x, j_y, j_z, P)$ selects the subset of points $P_j \subseteq P$ within the voxel indexed by $(j_x, j_y, j_z)$,

- $l(P_j)$ maps the selected point subset $P_j$ to a feature vector representing the voxel at $(j_x, j_y, j_z)$.

Substitute $P = gP$ into $\nu(P)$. Using the definition, we have:

$$\nu(gP)(j_x, j_y, j_z) = l\left(m(j_x, j_y, j_z, gP)\right).$$

The point set selection function $m$ is equivariant to group transformations, so we have:

$$m(j_x, j_y, j_z, gP) = g\,m(g^{-1}(j_x, j_y, j_z), P).$$

Substitute this into the expression for $\nu(gP)$:

$$\nu(gP)(j_x, j_y, j_z) = l\left(g\,m(g^{-1}(j_x, j_y, j_z), P)\right).$$

The PointNet $l$ is $SO(2)$-equivariant and $T(3)$-invariant, meaning:

$$l(gP_j) = \rho(\theta)l(P_j),$$

where $P_j = m(j_x, j_y, j_z, P)$, and $\rho(\theta)$ is the linear representation of the rotation group action.

Applying this to $l(g\,m(g^{-1}(j_x, j_y, j_z), P))$:

$$\nu(gP)(j_x, j_y, j_z) = \rho(\theta)l\left(m(g^{-1}(j_x, j_y, j_z), P)\right).$$

From the definition of $\nu(P)$, we know:

$$l\left(m(g^{-1}(j_x, j_y, j_z), P)\right) = \nu(P)(g^{-1}(j_x, j_y, j_z)).$$

Substituting this into the equation:

$$\nu(gP)(j_x, j_y, j_z) = \rho(g)\nu(P)(g^{-1}(j_x, j_y, j_z)).$$

Since $\mathcal{V} = \nu(P)$ is a voxel grid (in function representation), the group action on $\nu$ is defined as:

$$(g\nu(P))(j_x, j_y, j_z) = \rho(g)\nu(P)(g^{-1}(j_x, j_y, j_z)).$$

Thus, we have:

$$\nu(gP) = g\nu(P).$$

$\square$

# D. Additional Background of Group Symmetry

The group $T(3) \times SO(2)$ can be naturally decomposed into translation $T(3)$, which is handled using methods like 3D convolution, and rotation $SO(2)$, which is addressed via network design by introducing equivariant layers (Cesa et al., 2022) that respect $SO(2)$ transformations through appropriate representations of $SO(2)$ or its subgroups.

## D.1. Group Action of $SO(2)$

We focus on three particular representations of $g \in SO(2)$ or its subgroup $g \in C_u$ (containing $u$ discrete rotations) that define how the group acts on different data. Specifically:

Trivial Representation $\rho_0$: The trivial representation $\rho_0$ characterizes the action of $SO(2)$ or $C_u$ on an invariant scalar $x \in \mathbb{R}$ such that $\rho_0(g)x = x$. This means that every group element $g$ leaves the scalar $x$ unchanged.

Standard Representation $\rho_1$: The standard representation $\rho_1$ defines how $SO(2)$ or $C_u$ acts on a vector $v \in \mathbb{R}^2$ using a $2 \times 2$ rotation matrix. The action is given by $\rho_\omega(g)v = \left( \begin{smallmatrix} \cos g & -\sin g \\ \sin g & \cos g \end{smallmatrix} \right) v$. When $\omega = 1$, the representation $\rho_1(g)$ corresponds to the standard $2 \times 2$ rotation matrix.

Regular Representation $\rho_{\text{reg}}$: The regular representation $\rho_{\text{reg}}$ describes the action of $C_u$ on a vector $x \in \mathbb{R}^u$ via $u \times u$ permutation matrices. Let $g = r^m$ be an element of the cyclic group $C_u = \{1, r^1, \ldots, r^{u-1}\}$, and let $x = (x_1, x_2, \ldots, x_u) \in \mathbb{R}^u$. Then the action is defined by $\rho_{\text{reg}}(g)x = (x_{u-m+1}, x_{u-m+2}, \ldots, x_u, x_1, x_2, \ldots, x_{u-m})$. This operation cyclically permutes the coordinates of $x$ in $\mathbb{R}^u$.

A representation $\rho$ can also be constructed as a combination of different representations. Specifically, $\rho$ is defined as the direct sum $\rho = \rho_0^{n_0} \oplus \rho_1^{n_1} \oplus \rho_2^{n_2}$, which belongs to the general linear group $GL(n_0 + 2n_1 + 2n_2)$. In this case, $\rho(g)$ is a block diagonal matrix of size $(n_0 + 2n_1 + 2n_2) \times (n_0 + 2n_1 + 2n_2)$ that acts on vectors $x \in \mathbb{R}^{n_0 + 2n_1 + 2n_2}$.

## D.2. Group Action of $T(3)$

Follow the definition of + - The group $T(3)$ of 3D translations is an additive group, whose action is defined by shifting spatial coordinates. For example, for a point cloud $P = \{p_1, p_2, \ldots\}$ where $p_i = (x_i, y_i, z_i)$, the action of $g \in T(3)$ is $t \cdot p_i = (x_i + t_x, y_i + t_y, z_i + t_z)$. Similarly, for voxel-based representations, $T(3)$ acts by shifting the spatial indices of the voxel grid. Convolutions are inherently translationally invariant. ¡- I would say this Translation symmetry is naturally handled by operations such as 3D convolutions, which are inherently translation-equivariant.

# E. Training Detail

In the simulation experiments, we we use a batch size of 16 for training. Specifically, the observation contains one step of history observation, and 3 steps of history action and the output of the denoising process is a sequence of 18 action steps. In close-loop control we use all 18 steps for training and execute 18 steps, similar to prior work ((Xian et al., 2023)). In close-loop control 18 steps and 9 steps are used for training and execution, similar to setting of (Wang et al., 2024) a. We train our models with the AdamW ((Loshchilov & Hutter, 2019)) optimizer (with a learning rate of $10^{-4}$ and weight decay of $5*10^{-4}$). We use DDPM ((Ho et al., 2020)) with 100 denoising steps for both training and evaluation. We training each tasks with 100000 iterates.

# F. Detail of Simulation Tasks

Here are descriptions of 30 tasks, as shown in Figure 9, mentioned in simulation experiment:

1. **Pick/Lift**: Grasp and lift a block from the table.

2. **Push Button**: Press a button.

3. **Knife on Board**: Place a knife onto a cutting board.

4. **Put Money**: Put dollars in safe.

5. **Reach Target**: Move the gripper to a specified target location.

6. **Slide Block**: Slide a block across the table to certain area.

7. **Stack Wine**: Put wine bottles into a shelf.

8. **Take Money**: Take dollars from safe.

9. **Take Umbrella**: Retrieve an umbrella from a stand.

10. **Pick up Cup**: Grasp and lift a cup.

11. **Unplug Charger**: Disconnect a charger from an outlet.

12. **Close Door**: Shut a door fully.

13. **Open Box**: Lift the lid of a box.

14. **Open Fridge**: Pull the fridge door open.

15. **Frame off Hanger**: Remove a frame from a hanger.

16. **Open Oven**: Open the oven door.

17. **Books on Shelf**: Put book on a shelf.

18. **Wipe Desk**: Wipe a desk surface clean using a cloth.

19. **Cup in Cabinet**: Place a cup inside a cabinet.

20. **Shoe out of Box**: Remove a shoe from its box.

21. **Open Microwave**: Open a microwave door.

22. **Turn on Lamp**: Activate a lamp using its switch.

23. **Open Grill**: Lift the lid of a grill.

24. **Stack Blocks**: Stack blocks on top of each other.

25. **Stack Cups**: Arrange cups in a stacked configuration.

26. **Push 3 Buttons**: Press three buttons in a specific sequence.

27. **Plug USB in Computer**: Insert a USB device into a port.

28. **Open Drawer**: Pull a drawer open.

29. **Put Item in Drawer**: Pull a drawer open and place an object inside a drawer.

30. **Sort Shape**: Put shape in a shape sorter

## G. Real-World Experimental Settings

Our real-world experiments are conducted on a UR5e robotic arm equipped with a Robotiq 2F-85 gripper and three Intel RealSense D455 cameras as shown in Figure 10 . Demonstrations are collected using a 6-DoF 3DConnexion SpaceMouse at a 10 Hz rate, logging both the visual observations (from all three cameras) and the robot's end-effector actions (position, orientation, and gripper states).

## H. Comparison With Hierarchical Diffusion Policy(HDP)

We also compare our policy with another hierarchical baseline, HDP (Ma et al., 2024), by selecting seven tasks from the HDP paper that we evaluated. We then compare the success rates on these tasks, as shown in Table 6. Our approach achieves an absolute mean improvement of 20%, demonstrating superior sampling efficiency.

*Table 6.* **Performance of HDP and HEP on 7 Tasks.**

| Method(**Open-loop**) | **Mean** | Reach Target | Pick Up Cup | Open Box | Open Drawer | Open Microwave | Open Oven | Knife on Board |
|---|---|---|---|---|---|---|---|---|
| HDP | 74 | **100** | 82 | 90 | 90 | 26 | 58 | 72 |
| HEP (Ours) | **94**(+20) | **100** | **98**(+16) | **100**(+10) | **94**(+4) | **82**(+56) | **87**(+29) | **96**(+24) |

*Table 7.* **Performance of Different Ablations on Various Tasks.**

| Method | Mean | Lamp on | Open microw. | Push 3 buttons | Push button | Open box | Insert USB |
|---|---|---|---|---|---|---|---|
| No Hierarchy | 0.51 | 0.28 | 0.42 | 0.01 | 0.96 | 0.99 | 0.38 |
| No Equi No FT | 0.60 | 0.21 | 0.44 | 0.53 | 0.96 | 0.99 | 0.51 |
| No Equi | 0.70 | 0.41 | 0.53 | 0.67 | 0.98 | 0.99 | 0.64 |
| No FT | 0.78 | 0.75 | 0.56 | 0.73 | 0.98 | 0.99 | 0.68 |
| No Stacked Voxel | 0.84 | 0.77 | 0.65 | 0.87 | 0.99 | 0.99 | 0.79 |
| **Complete Model** | **0.94** | **0.95** | **0.82** | **0.99** | **1.00** | **1.00** | **0.90** |

## I. Full Result of Ablation Study

We show the full result of ablation study (subsection 5.4) here at Table 7.

## J. Voxelization Details

We build our voxelization function based on (Contributors, 2020). The size of our voxel grid is 64*64*64 with maximum 6 points within it.

## K. Equivariance Error

We conducted an experiment measuring the equivariance error specifically on $C_4$, a subgroup of $SO(2)$. This allowed us to quantify the difference between the rotated output and the output from a rotated input. The experimental results are summarized in the Table 8.

*Table 8.* **Equivariance Error Under Different Rotations**

| Rotation | Equivariance Error |
|---|---|
| $0°$ | $0\%$ |
| $90°$ | $0.013\%$ |
| $180°$ | $0.006\%$ |
| $270°$ | $0.009\%$ |

## L. Performance Under Human Perturbation

We conducted an extra experiment evaluating the success rate of executing a block stacking task under human perturbation mimicking a dynamic environment and include the results in Table 9.

*Table 9.* **Success Rate Under Human Perturbation**

| Task | Success Rate |
|---|---|
| Blocks stacking | 0.8 |

*Figure 9.* **All simulation tasks we evaluate on**

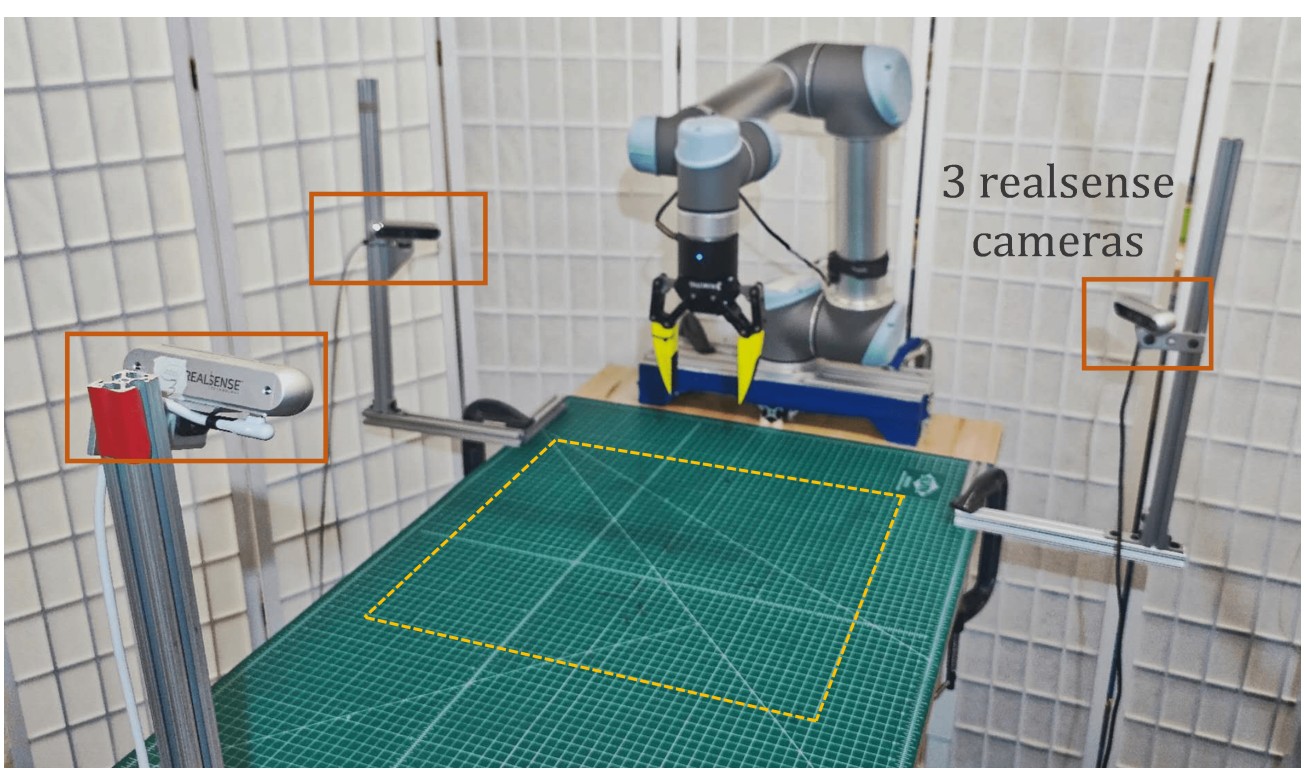

*Figure 10.* **Real-world Experiment Setting**

