# OpenReview forum: "Hierarchical Equivariant Policy via Frame Transfer"
_ICML.cc/2025/Conference — ICML 2025 poster_

### Official Review · Reviewer_1oMk · 2025-03-11

**Overall Recommendation:** 3

**Summary:**

Hierarchical Equivariant Policy (HEP) enhances hierarchical policy learning by introducing a frame transfer interface, where the high-level agent’s output serves as a coordinate frame for the low-level agent, improving flexibility and inductive bias. It also integrates domain symmetries at both levels, ensuring overall equivariance. HEP achieves state-of-the-art performance in complex robotic manipulation tasks, demonstrating improvements in both simulation and real-world settings.

**Claims And Evidence:**

The high-level policy outputs only x, y, z positions, while the low-level policy generates concrete actions. This approach is commonly used in hierarchical planning, where the high-level policy produces key points and the low-level policy generates trajectories. Additionally, the frame transfer mechanism resembles a coordinate transformation to a target-frame-centric representation. The authors should provide more explanation to clarify the novelty of their approach.

**Essential References Not Discussed:**

N/A

**Experimental Designs Or Analyses:**

All the baselines use 3D scene representations; a comparison with state-of-the-art image-based visuomotor policies would provide a more comprehensive evaluation.

**Methods And Evaluation Criteria:**

Can the method be applied in scenarios with only image input, without requiring RGB-D inputs?

During the evaluation, is a separate policy trained for each task? Also, does the policy incorporate language inputs?

How is the Equivariant Diffusion Policy used? Does its sampling procedure differ from that of a standard Diffusion Policy?

**Other Comments Or Suggestions:**

N/A

**Other Strengths And Weaknesses:**

N/A

**Questions For Authors:**

N/A

**Relation To Broader Scientific Literature:**

N/A

**Theoretical Claims:**

N/A

---

> ### Author Rebuttal · Authors · 2025-04-01
>
> We thank the reviewer for their response. We respond below:
>
> ---
> > “...The authors should provide more explanation to clarify the novelty of their approach.”
>
> We agree with the reviewer that there exist various works on hierarchical policy learning. However, our method is the first to incorporate equivariant learning into hierarchical policy. This is both novel, and leads to sizable performance increases.  Additionally, our proposed frame transfer interface ensures translational equivariance and imposes soft constraints on the low-level agent.
>
> We refer Reviewer 4 to Reviewer 1’s assessment of our method's novelty
> > “Can the method be applied in scenarios with only image input, without requiring RGB-D inputs?”
>
>
>
> Currently, our approach requires a 3D understanding of the environment, which we obtain using RGB-D inputs. Given the expense and inherent noise associated with RGB-D sensors, an interesting direction for future work would be to extend our framework to scenarios that rely solely on RGB images. Recent advances in depth-estimation foundation models [1,2] may enable the generation of accurate 3D representations from pure image data. Additionally, in tabletop settings where a top-down camera is available, pixel coordinates can be directly mapped to spatial (x,y) positions without requiring explicit depth measurements, thus potentially enabling SE(2) equivariance in the policy.
>
> [1] Guo, Y, et al. "Depth Anything: Unleashing the Power of Large-Scale Image-Text Pretraining for Zero-Shot Depth Estimation." arXiv preprint arXiv:2311.16502, 2023.
>
> [2] Wen, B, et al. "FoundationStereo: Zero-Shot Stereo Matching." arXiv preprint arXiv:2501.09898, 2025.
>
> > “...separate policy trained for each task? ...does the policy incorporate language?”
>
> We are evaluating our policy under a single-task setting following former work [3] for fair comparison. However, it should be straightforward to operate in a multitask setting by adding a language feature to the observation, as is done in [4,5]. Evaluating the performance in multi-task settings can be an interesting direction for future work.
>
> [3] Xian, Z et al "ChainedDiffuser: Unifying Trajectory Diffusion and Keypose Prediction for Robotic Manipulation." CoRL. PMLR, 2023.
>
> [4] Ke, J, et al. "3D Diffuser Actor: Policy Diffusion with 3D Scene Representations." CoRL. PMLR, 2024.
>
> [5] Shridhar, M, et al. "Perceiver-Actor: A Multi-Task Transformer for Robotic Manipulation." CoRL. PMLR, 2023.
>
> ---
>
> > “How is the Equivariant Diffusion Policy used?...sampling procedure differ from that of a standard Diffusion Policy?”
>
> Our method improves upon standard diffusion policy in two key ways. Specifically, our policy is explicitly equivariant [6], meaning that it is automatically able to generalize to new problem instances related by symmetry. This is not inherent in Diffusion Policy [7]. Secondly, our method utilizes a hierarchical decomposition — namely, a high-level and low-level agent — not found in standard diffusion policy. As shown in our experiment result in Table 1 in our response to reviewer 3 cFSf (referenced here due to character limitations) and Table 1 here, these two changes enable our method to significantly outperform standard diffusion policy.
>
> [6] Weiler, M. et al. "E(2)-Equivariant Neural Networks for Image Classification and Beyond." ICLR, 2019.
>
> [7] Chi,C et al. "Diffusion Policy: Visuomotor Policy Learning via Action Diffusion." RSS, 2023.
>
> ---
>
> > “...baselines use 3D scene representations; a comparison with state-of-the-art image-based visuomotor policies would provide a more comprehensive evaluation.”
>
> As shown in the prior work [9], image-based Diffusion Policy [7] performs poorly in RLBench tasks, where the 3d representation of the workspace is essential. We also performed an extra experiment comparing the image-based diffusion policy against our method and we report the result here:
>
> ### Table 1. Performance on 5 Tasks from RLBench
> | Method (Closed-loop) | Mean | Turn On Lamp | Open Microwave | Push 3 Buttons | Open Drawer | Put Item in Drawer |
> |-|-|-|-|-|-|-|
> | **Ours** | **79** (+23) | **60** (+32) | **64** (+22) | **37** (+36) | **95** (+41) | **76** (+28)  	|
> | EquiDiff [8] | 57 | 28 | 42 | 1| 54|48|
> | DiffPo (Img) [7] | 2.8  | 4| 1 | 0  | 7| 2  |
>
> We follow the hyperparameters from the original work, with the exception that we use RGB images from the same four cameras used in our setup (wrist, front, right shoulder, and left shoulder) and at the same resolution, to ensure a fair comparison.
>
>
> [8] Wang ,D ,et al "Equivariant Diffusion Policy." CoRL. PMLR, 2024.
>
> [9] Shridhar, M et al "Generative Image as Action Models." CoRL PMLR, 2024.

---

### Official Review · Reviewer_cFSf · 2025-03-14

**Overall Recommendation:** 3

**Summary:**

The paper proposes to develop SO(2)xT(3) equivariant hierarchical policy for imitation learning learnt policy. The high-level proposes the translation target in the canonical world frame. While the low-level policy uses the local frame to diffuse the action. The equivariance is achieved via frame transfer interface and voxel repr. Experiments show that the proposed hierarchical policy achieves ~10% performance when the number of demonstration is very limited.

**Claims And Evidence:**

The claim is well supported by the experiments. Furthermore, i agree that with these design choices, the policy is indeed equivariant.

**Essential References Not Discussed:**

There should be work on equivariant policies, e.g., Equivariant Diffusion Policy, Wang et al.

**Experimental Designs Or Analyses:**

The experiment is comprehensive. However, i think related baselines are missing like equivariant policies, e.g., Equivariant Diffusion Policy. I would like understand the difference of having a hierarchical policy and more simple versions.

**Methods And Evaluation Criteria:**

As RL-Bench is the standard to test imitation learning algorithms, i believe the result is quite convincing on its capability to generalize well in the low data domain.

**Other Comments Or Suggestions:**

None

**Other Strengths And Weaknesses:**

The design of the hierarchical policy is reasonable for the equivariant policy. However, i think evaluation is needed to show that a hierarchical policy is indeed needed for solving these imitation learning tasks.

**Questions For Authors:**

1. Could you show results of other equivariant policies?

**Relation To Broader Scientific Literature:**

The work studies methods that can allow the imitation policy works well in low data domain. This is an important problem for robot learning.

**Theoretical Claims:**

I agree that the theoretical claims on the equivariance property is correct.

---

> ### Author Rebuttal · Authors · 2025-04-01
>
> We thank the reviewer for their response. We respond below:
>
> > "...related baselines are missing like equivariant policies, e.g., Equivariant Diffusion Policy. I would like understand the difference of having a hierarchical policy and more simple versions"
>
> > "There should be work on equivariant policies, e.g., Equivariant Diffusion Policy, Wang et al."
>
>
> > "Could you show results of other equivariant policies?"
>
> Thank you for bringing this to our attention. We did provide comparisons with equivariant policies, specifically the Equivariant Diffusion Policy [1], labeled as **"EDP"** in Table 1. However, we acknowledge that this reference may not have been sufficiently clear. In our revised manuscript, we will explicitly clarify that **"EDP"** refers to the Equivariant Diffusion Policy and emphasize this comparison more clearly to avoid confusion.
>
> Additionally, we've conducted an ablation study on the hierarchical architecture to better understand its benefits:
>
> ### Table 1. Revised Ablation Experiment
>
> | **Method**     	| **Mean** | **Lamp on** | **Open microw.** | **Push 3 buttons** | **Push button** | **Open box** | **Insert USB** |
> |--------------------|---------:|------------:|-----------------:|-------------------:|----------------:|-------------:|---------------:|
> | No Hierarchy   	| 0.51 	| 0.28    	| 0.42         	| 0.01           	| 0.96        	| 0.99     	| 0.38       	|
> | No Equi No FT  	| 0.60 	| 0.21    	| 0.44         	| 0.53           	| 0.96        	| 0.99     	| 0.51       	|
> | No Equi        	| 0.70 	| 0.41    	| 0.53         	| 0.67           	| 0.98        	| 0.99     	| 0.64       	|
> | No FT          	| 0.78 	| 0.75    	| 0.56         	| 0.73           	| 0.98        	| 0.99     	| 0.68       	|
> | No Stacked Voxel   | 0.84 	| 0.77    	| 0.65         	| 0.87           	| 0.99        	| 0.99     	| 0.79       	|
> | **Complete Model** | **0.94** | **0.95**	| **0.82**     	| **0.99**       	| **1.00**    	| **1.00** 	| **0.90**   	|
>
> Our findings demonstrate that incorporating a hierarchical structure notably enhances performance, particularly on long-horizon tasks.
>
> ---
>
> **[1]** Wang, Dian, Stephen Hart, David Surovik, Tarik Kelestemur, Haojie Huang, Haibo Zhao, Mark Yeatman, Jiuguang Wang, Robin Walters, and Robert Platt. *"Equivariant Diffusion Policy."* Conference on Robot Learning (CoRL). PMLR, 2024.

---

### Official Review · Reviewer_Vgsm · 2025-03-14

**Overall Recommendation:** 4

**Summary:**

The paper proposes a novel Hierarchical Equivariant Policy (HEP) framework for robotic manipulation tasks, combining hierarchical learning with translation and rotation equivariance via a flexible Frame Transfer interface. HEP decomposes tasks into high-level coarse predictions and low-level fine-grained trajectory generation, improving flexibility, sample efficiency, and spatial generalization. Experiments show HEP significantly outperforms existing approaches in simulation and real-world robotic tasks, especially those demanding precise control and long-horizon planning.

**Claims And Evidence:**

The experiments support the main claims. The main claims are:
- HEP significantly improves robotic manipulation performance in both simulation (RLBench tasks) and real-world experiments compared to state-of-the-art methods.
- Frame Transfer Interface effectively provides flexibility and efficiency in hierarchical policy learning.

Empirical evaluations across 30 diverse tasks demonstrate that HEP achieves higher success rates compared to baseline methods.

**Essential References Not Discussed:**

Nothing I can think of.

**Experimental Designs Or Analyses:**

In open-loop training, the low-level target is constructed by interpolating between the consecutive keyframes. This interpolation may fail in a clustered environment due to the need for obstacle avoidance. This also introduces one of the main limitations of the experimental setup: the method is tested only on a simple, unclustered table-top setting.

**Methods And Evaluation Criteria:**

The proposed methods and simulation benchmark make sense for the problem. The inclusion of real-world experiments is also a plus. The ablations demonstrate the value of different components. One-shot generalization tests are also interesting.

**Other Comments Or Suggestions:**

N/A

**Other Strengths And Weaknesses:**

N/A

**Questions For Authors:**

N/A

**Relation To Broader Scientific Literature:**

The paper’s contributions are well-motivated, and address important limitations of existing approaches.

**Theoretical Claims:**

The theoretical claims seem correct under the assumptions stated, although they are impossible to verify due to the deep neural net.

---

> ### Author Rebuttal · Authors · 2025-04-01
>
> We thank the reviewer for their response. We respond below:
> > “The theoretical claims seem correct under the assumptions stated, although they are impossible to verify due to the deep neural net.”
>
> We thank the reviewer for raising this important point. While our theoretical claims appear valid under the stated assumptions, we acknowledge the difficulty of directly verifying these assumptions due to the complexity of deep neural networks. To address this concern empirically, we conducted an experiment measuring the equivariance error specifically on C4, a subgroup of SO(2). This allowed us to quantify the difference between the rotated output and the output from a rotated input. The experimental results are summarized in the table below:
>
> **Table 1. Equivariance Error Under Different Rotations**
>
> | Rotation | Equivariance Error |
> |----------|--------------------|
> | 0°   	| 0%             	|
> | 90°  	| 0.013%         	|
> | 180° 	| 0.006%         	|
> | 270° 	| 0.009%         	|
>
> As you can see, the network is equivariant, modulo numerical error, to rotational transformation. Also, our network inherits the transnational equivariance through the Unet and frame transfer architecture.
>
> ---
>
> > “In open-loop training, the low-level target is constructed by interpolating between the consecutive keyframes. This interpolation may fail in a clustered environment due to the need for obstacle avoidance. This also introduces one of the main limitations of the experimental setup: the method is tested only on a simple, unclustered table-top setting.”
>
> We thank the reviewer for the question. In Section 6 **“Robust to Environment Variations”**, we demonstrate that the trained policy would be robust to some disturbance objects in the workspace. We conducted an extra experiment evaluating the success rate of executing a block stacking task under human perturbation mimicking a dynamic environment and include the results here:
>
> **Table 2. Success Rate Under Human Perturbation**
>
> | Task        	| Success Rate |
> |------------------|--------------|
> | Blocks stacking  | 0.8      	|
>
> However, the reviewer makes a good point that if the demonstration data does not contain obstacle avoidance while an obstacle is introduced at test time, the trained policy might fail. In future work, we propose to add an additional hierarchy of a trajectory optimization layer, which would refine the policy for obstacle avoidance.

---

### Official Review · Reviewer_Sk4u · 2025-03-14

**Overall Recommendation:** 4

**Summary:**

The paper introduces Hierarchical Equivariant Policy (HEP), a framework for hierarchical reinforcement learning that integrates equivariance (geometric symmetry) into a two-level policy architecture. The high-level policy outputs a coarse 3D subgoal (keypose), essentially a translation in space representing the next target location in the task. This subgoal is then used to “frame-shift” the low-level policy’s coordinate system via a Frame Transfer interface, meaning the low-level agent perceives the world relative to the high-level’s suggested frame.

**Claims And Evidence:**

The paper introduces a novel integration of equivariance into hierarchical policy learning. This is the first work to imbue a two-level (coarse-to-fine) policy with symmetry properties, theoretically proving that the combined high-low policy remains equivariant to translations and rotations. This theoretical contribution is non-trivial and extends prior equivariant RL approaches (which were single-level) to a hierarchical setting. By decomposing the symmetry into a global part (at the high level) and a local part (at the low level), the design cleverly ensures the overall policy is invariant to spatial transformations of the task, which is a new insight with potentially broad impact on how we design policies that generalize across space and orientation. Instead of the high-level simply commanding a fixed goal state for the low-level (as in prior two-stage approaches), it outputs a coordinate frame shift (a 3D translation) that serves as a context for the low-level. This provides a strong inductive bias (the low-level is “anchored” to work towards the subgoal) without rigidly constraining the low-level’s behavior. The low-level agent can thus refine the trajectory locally, handling details and adjustments that the high-level might miss. This hierarchical decomposition bridges the gap between keyframe-based and trajectory-based learning methods  – combining their advantages.

**Essential References Not Discussed:**

Shao, Jianzhun, et al. "Pfrl: Pose-free reinforcement learning for 6d pose estimation." Proceedings of the IEEE/CVF Conference on Computer Vision and Pattern Recognition. 2020.

**Experimental Designs Or Analyses:**

To efficiently handle visual input, HEP uses a stacked voxel grid representation of point clouds (from multi-view RGB-D cameras) processed by a 3D equivariant U-Net, enabling rich 3D features and fast inference. On the experimental side, the paper provides an extensive evaluation of HEP on 30 robotic manipulation tasks from RLBench and several real-world robot tasks.

**Methods And Evaluation Criteria:**

HEP demonstrates state-of-the-art results on a wide range of tasks. It consistently outperforms multiple baseline methods, including advanced diffusion-based policies, by a significant margin. A notable strength is the method’s demonstrated ability to generalize beyond its training conditions. The one-shot learning experiment, where HEP learned a task from a single demo and still succeeded 80% of the time on novel object configurations, is a compelling result. This level of generalization and robustness is a major practical strength, as real-world deployments often face variations that are not seen in training. The paper goes beyond simulation and validates the approach on a real robotic system, which strengthens the work significantly. Additionally, the authors include ablation studies that clearly quantify the contribution of each component: removing equivariance, Frame Transfer, or the voxel encoder each drops performance significantly (e.g. removing equivariance alone reduces success by 24%).

**Other Comments Or Suggestions:**

No

**Other Strengths And Weaknesses:**

•	While HEP addresses translational equivariance thoroughly, the high-level Frame Transfer interface handles only translations (T(3)) and not rotations of the coordinate frame. The equivariance to rotations is incorporated in the network (policy) architecture for planar rotations (SO(2)), but the high-level does not explicitly output a desired orientation for the subgoal. This is noted by the authors as a limitation: currently the framework isn’t equipped to specify or leverage rotations in the subgoal, which could limit performance on tasks where orientation of the end-effector is critical. The paper suggests that extending Frame Transfer to include rotation is future work. Until that is realized, the method might struggle or require ad-hoc solutions in scenarios like screwing in a lightbulb or opening a door, where the goal orientation matters as much as position.

•	Another limitation is that the experiments are confined to tabletop robotic manipulation tasks. All tasks in the paper involve a fixed robotic arm interacting with objects on a table or in a small workspace. This leaves open the question of how well the approach scales to other domains (e.g., legged locomotion, navigation, or deformable object manipulation). The authors explicitly note that extending HEP to more complex robots like humanoids is a promising direction, but it remains to be seen if the current design would directly apply or if significant modifications are needed. For example, a humanoid might require a deeper hierarchy or different equivariances. Thus, the generality of HEP beyond manipulation isn’t proven.

•	HEP is trained via behavior cloning on demonstration data. While the approach is more sample-efficient than prior methods it still fundamentally requires demonstration trajectories for each new task. This reliance on expert demos could be a bottleneck in scenarios where obtaining demonstrations is time-consuming or costly. As a minor point, hyperparameter sensitivity (e.g., how the choice of the interval m for high-level steps, or the number of diffusion steps, affects performance) is not deeply discussed – presumably these were tuned on a subset of tasks, but it’s not reported, leaving a bit of uncertainty on how robust the training setup is.

•	While the related work is thorough, there might be recent works on hierarchical RL or imitation (outside of diffusion models) that could be acknowledged for completeness. These are relatively minor and did not significantly hinder understanding or merit of the work.

**Questions For Authors:**

1.	HEP showed strong results even with very few demonstrations in some cases. Could the authors elaborate on what aspects of HEP enable such one-shot generalization? For example, is it largely the equivariant inductive bias that makes one demo cover many situations via symmetry?

2.	how much does the hierarchical decomposition alone (even without equivariance) contribute versus a flat equivariant policy?

3.	In real-world tests some failures were due to misalignment from depth sensor noise. How might the system be made more robust to such errors?

**Relation To Broader Scientific Literature:**

The paper is related to existing hierarchical methods in robotic manipulation such as Ma et al., 2024; Xian et al., 2023. This paper proposes an approach where the high-level agent predicts a keypose in the form of a coarse 3D location representing a subgoal of the task. This location is then used to construct the coordinate frame for the low-level policy.

**Theoretical Claims:**

The low-level policy predicts a sequence of fine-grained actions (trajectory) relative to the subgoal frame instead of absolute coordinates. This design preserves a strong inductive bias (anchoring the low-level to an intermediate goal) while allowing flexibility for the low-level to adjust and refine the trajectory locally. Crucially, the authors incorporate domain symmetries into both levels of the policy: the high-level subgoal selection and the low-level trajectory generation are designed to be equivariant to translations and in-plane rotations (T(3) × SO(2) symmetry). This means if the environment or task is shifted or rotated, the HEP’s policy will produce a correspondingly shifted/rotated action sequence, greatly enhancing spatial generalization. Theoretical propositions (with proofs in the appendix) show that under the given design, the entire hierarchical policy is equivariant to those transformations.

---

> ### Author Rebuttal · Authors · 2025-04-01
>
> We thank the reviewer for their thoughtful feedback. We respond below:
>
> ---
>
> > "HEP addresses translational equivariance thoroughly... currently, the framework isn’t equipped to specify or leverage rotations... could limit performance on tasks where orientation is critical."
>
> Thank you for bringing this up! We agree that predicting orientation subgoals at the high level could potentially further improve sample efficiency. However, we would like to clarify that our current low-level agent already predicts SE(3) actions, enabling it to handle tasks requiring complex end-effector orientations, such as door closing, fridge opening, and pot cleaning. That being said, we are actively working on an extension of this paper where the high-level explicitly predicts goal orientations. We will thoroughly investigate and compare the advantages and disadvantages of these two orientation-handling strategies in our follow-up work.
>
> > "Experiments... confined to tabletop manipulation... scalability to other domains remains unclear."
>
> We appreciate the reviewer's insightful suggestion. Extending HEP to more complex robots, such as humanoids, is indeed a valuable direction for future research. We recognize that the generality of HEP in cross-embodied scenarios remains to be fully established. Nonetheless, we firmly believe that HEP effectively demonstrates two fundamental principles essential for efficient policy learning: special Euclidean group symmetry and hierarchical policy decomposition. These principles are expected to inform and inspire future developments across various robotic domains.
>
> > "HEP relies on demonstrations... could be a bottleneck... hyperparameter sensitivity was not thoroughly discussed."
>
> We appreciate the reviewer's valuable insight. We agree that dependence on expert demonstrations can be a limiting factor, especially given the time-consuming nature of obtaining demonstrations. While this study primarily explores behavior cloning, we believe future research could effectively extend HEP into reinforcement learning frameworks, thus eliminating the necessity for human demonstrations.
> Regarding hyperparameter sensitivity, we would like to clarify that our approach did not involve any hyperparameter tuning. Specifically, we directly adopted the high-level hyperparameters from Act3D and the low-level hyperparameters from Equivariant Diffusion Policy without any finetuning. We believe this choice underscores the robustness of our method. We will also make this explicit in the final version of the manuscript.
>
>
> > "Related work is comprehensive... recent hierarchical RL or imitation learning works outside diffusion models should be acknowledged."
>
> Thanks for your valuable advice. We added following works as our related works:
>
> [1] Wang, C, et al. "MimicPlay: Long-Horizon Imitation Learning by Watching Human Play." CoRL. PMLR, 2023.
>
> [2] Luo, J, et al. "Multi-Stage Cable Routing through Hierarchical Imitation Learning." IEEE Transactions on Robotics, 2024.
>
> [3] Triantafyllidis, E, et al. "Hybrid Hierarchical Learning for Solving Complex Sequential Tasks using the Robotic Manipulation Network (ROMAN)." Nature Machine Intelligence, 2023.
>
> [4] Bagaria, A, et al. "Effectively Learning Initiation Sets in Hierarchical Reinforcement Learning." NeurIPS, 2023.
>
> [5] Shao, J, et al. "Pfrl: Pose-free reinforcement learning for 6d pose estimation." CVPR. 2020
>
> ---
>
> ### Reviewer’s Questions:
>
> > "HEP showed strong one-shot generalization... Is it largely due to equivariant inductive bias?"
>
> Thank you for bring this up. We believe the reason is that we use a 3D Unet as our high level (which is T(3) equivariant), but most importantly, our frame transfer interface passes the equivariance and generalization ability to our low level, leading to an improved generalization ability of the whole policy.
>
> > "Contribution of hierarchical decomposition alone (without equivariance) compared to flat equivariant policy?"
>
> This is a great question. We conducted an ablation study with hierarchical decomposition. And we put it into our response to Reviewer 3 cFSf due to character limitation. Please see Table 1 in our response to reviewer 3 cFSf for more details. As shown in the table, removing the hierarchical decomposition lead to a significant drop in performance, demonstrating importance of a hierarchical structure especially on long horizon task.
>
> > "Improving robustness to depth sensor noise?"
>
> We thank the reviewer for raising this important point. While the high sample efficiency of our policy allows us to train directly on real-world data—enabling the policy to adapt to sensor noise to some degree—we agree that introducing additional noise during training (e.g., applying a dropout layer to randomly remove points) could further enhance the robustness of the system to sensor noise. We will explore this idea in future experiments.
>
> Thank you again for your insightful feedback!

---

### Decision · Program_Chairs · 2025-05-01

**Decision:**

Accept (poster)

**Comment:**

This submission proposes the Hierarchical Equivariant Policy (HEP) framework, introducing equivariance into hierarchical imitation learning via a novel Frame Transfer interface. HEP demonstrates strong empirical performance across 30 RLBench tasks and real-world robotic manipulation scenarios, outperforming several baselines including diffusion-based and equivariant methods. The reviewers agreed that the submission presented the clear theoretical formulation, extensive experiments, one-shot generalization results, and real-robot deployment.

All reviewers agree that this submission is a well-executed contribution, particularly in integrating symmetry properties into multi-level policy structures. They also appreciated the robustness to data scarcity, the clean decomposition into high- and low-level behaviors, and strong empirical validation with meaningful ablations. While there were a few concerns (handling translational equivariance primarily and limited evaluation results), the rebuttal addressed most concerns (providing additional analysis, ablations, and clarification on experimental setups.)

Overall, the submission is solid, and the AC recommends acceptance. It is well-positioned to inspire future research extending hierarchical equivariant control. I recommend a poster, only because of slightly limited coverage of experiments in terms of scaling.